# A comparative study on immune responses to demineralized and decellularized bone substitute following intraperitoneal implantation in mouse model

K. G. Aghila Rani[1☺], Ali Al Qabbani[2,3☺], Sausan AlKawas[2], Manju Nidagodu Jayakumar[1,4], S. A. H. Suzina[5], A. R. Samsudin[2]*, Ahmad Azlina[3]*

1 Research Institute for Medical and Health Sciences, University of Sharjah, Sharjah, UAE, 2 Department of Oral and Craniofacial Health Sciences, College of Dental Medicine, University of Sharjah, Sharjah, UAE, 3 School of Dental Sciences, Universiti Sains Malaysia, Kota Bharu, Malaysia, 4 Department of Biotechnology, Birla Institute of Technology and Science (BITS Pilani), Dubai Campus, Academic City, Dubai, UAE, 5 Tissue Bank, School of Medical Sciences, Universiti Sains Malaysia, Kota Bharu, Malaysia

☺ These authors contributed equally to this work.
* drabrani@sharjah.ac.ae (ARS); azlinakb@usm.my (AA)

## Abstract

The immunological sensitization of implanted bone grafts is crucial for long-term success. This study aimed to investigate the immune responses following implantation of lyophilized demineralized (DMB) and lyophilized decellularized (DCC) bovine cancellous bone substitutes, respectively, in mouse models of peritoneal implantation to evaluate the effectiveness of DMB and DCC processing methods. The DMB and DCC substitutes were prepared using published methods. BALB/c mice were divided into four groups (n = 4). A small abdominal incision was created to deliver the DMB or DCC materials into the peritoneal cavity. The first group received native unprocessed bone, while the second group was sham-operated (SO). The third and fourth groups received DMB and DCC substitutes, respectively. The immunogenicity effects of the implants were assessed through WBC count, spleen index, CD4 + /CD8 + counts, cytokine expression, and histology analysis of the spleen, liver and kidney. Native controls displayed systemic inflammation. The DMB group showed an increased trend in WBC count, cytokine profile and spleen index on day seven, followed by a considerable reduction in the DCC group compared to DMB on days 14 and 21. The native group showed significantly higher CD4$^+$/CD8$^+$ T-cells and proinflammatory cytokines (IL-12, TNF-α, IFN-γ, MCP-1, IL-6). Additionally, the DMB group showed significantly higher mRNA levels for IL-1β, TNF-α, IL-6, and the anti-inflammatory cytokine IL-10. The DMB group further exhibited a significantly higher CD4$^+$ count, while the DCC group demonstrated higher CD8$^+$ T-cells on day 1. Histological assessments of the liver and kidney revealed pyknotic nuclei, necrotic cells, and extravasated RBCs in the native group and, to a lesser extent, in the DMB group,

**Data availability statement:** All relevant data are within the paper and its Supporting Information files.

**Funding:** We would also like to acknowledge the funding resources from the University of Sharjah, UAE (grant no. 2301100169) awarded to Prof. Rani Samsudin.

**Competing interests:** The authors have declared that no competing interests exist.

while the DCC group showed normal morphology similar to Sham. Both DMB and DCC demonstrated favourable immunocompatibility properties, while DCC exhibited further immune tolerance in the mouse model.

## Introduction

Processed bovine bone graft scaffolds have been in use for more than half a century, with varying degrees of success, and their popularity continues to grow [1–3]. The two main bovine bone substitute preparation procedures are demineralization and decellularization [4,5]. Demineralization and Decellularization preparations and their osteogenic evaluation *in vitro* and *in vivo* had been published previously [5,6]. Both processes lead to different physicochemical characteristics of the final form of bone substitute [7]. Physicochemical cues like topography, chemical composition and wettability can impact the inflammatory response and tissue integration after bovine bone implantation [8]. It is known that protein adsorption activity at the host-bone graft interphase is crucial for the determination of the immunogenic fate of the material [9]. Therefore, variations in biomaterial surface properties caused by different manufacturing processes influence protein and cell adhesion at the host-material interface, shaping immune responses that impact tissue integration and implant success.

Both innate and adaptive immune systems orchestrate the immune responses to allogenic and xenogeneic bone grafting [10]. Previously, biomaterial scientists focused on "biocompatibility" as a yardstick of material safety and implantation success [11]. The immunogenic responses of implantable biomaterials have been underexplored in the literature, primarily due to the complexity of immune interactions involving diverse cell types and intricate signaling pathways, making them challenging to study comprehensively. Furthermore, earlier perspectives on biomaterials did not fully recognize the importance of immunogenicity, contributing to its relative underrepresentation in past studies. These factors collectively explain the limited investigation in this area [12,13]. Given this knowledge gap, we investigated immune responses and the parameters influencing them to enhance our understanding of their impact on biomaterial integration and success.

Utilizing *in vivo* animal models instead of *in vitro* assays may yield a more precise assessment of immunological competence and immunotoxicity of the biomaterial. Preclinical immunological investigations of bone substitutes using rodents may help characterize innate and acquired immune systems. While the innate immune responses are often addressed, the cell-mediated host-tissue responses to natural ECM products could be characterized by analyzing the mononuclear cell infiltrates and T-cell responses, evidence of systemic inflammation, and immunological injury in immune-related organs, including the spleen, liver, and kidney [14].

This study aimed to investigate the immune responses of lyophilized demineralized (DMB) and lyophilized decellularized (DCC) bovine cancellous bone substitutes following implantation into the peritoneal cavity of BALB/c mice over a 21-day observation period.

## Materials and methods

### Preparation of demineralised (DMB) and decellularized (DCC) bovine bone substitutes

Fresh bovine femoral head bone was immediately obtained after the slaughter of a 24-month-old calf at a local abattoir (Sharjah Slaughterhouse, UAE) and subsequently sectioned into cubic blocks measuring 10 mm³. These bone blocks were divided into two groups and subjected to processing to produce lyophilized DMB and DCC bone substitutes. Our prior publications have comprehensively elucidated the processing techniques employed in this study according to previously established methods [6]. In brief, the processing protocol encompassed demineralization and decellularization steps, including initial cleaning via hydrostatic water pressure and treatment with a series of detergents, chemicals, and enzymes. Both the treated bone blocks were subsequently subjected to a freeze-drying process, after which the resulting DMB and DCC scaffolds were crushed into granules of 0.5mm in size and subjected to gamma irradiation at a dose of 25 kGy for utilization in the current *in vivo* experiments (S1 fig).

### Animals

This study was approved by the Animal Care and Use Committee of the University of Sharjah, Sharjah, UAE (ACUC-18-10-22-01). BALB/c inbred male mice, eight weeks old, weighing 23–24 g provided by the Animal Care facility of the University of Sharjah, UAE, were used in this study. All animals were given one week to adapt to their environment before undergoing experiments and were maintained on a 12-h-dark/12-h-light cycle at about 20°C and allowed free access to a pellet diet (Altromin, 1324-10 mm, Germany) and water during the experimental period. Animals were regularly monitored by an assigned veterinarian and the report was prepared according to the ARRIVE guidelines for reporting animal research [15]. All animals were individually housed in separate cages and administered a daily intramuscular injection of Bupresol at a dose of 20,000 U/kg following surgery to manage potential pain or discomfort. Animals were euthanized at the end of the experimental period following the guidelines set by Animal Care and Use Committee of the University of Sharjah. The method of sacrifice involved cervical dislocation ensuring minimal distress to the animals.

### Implantation surgery

A total of 64 mice were randomly divided into four experimental groups (n = 16). Intraperitoneal implantation surgery was performed based on a previous technique [16,17]. Before the treatment, animals were acclimatized for one week to adapt to the laboratory environment. The first group of animals received unprocessed native bone, the second group was sham-operated (SO), the third group received DMB, and the fourth group received DCC substitutes. During the implantation procedure, all animals were anesthetized using xylazine (20–80 mg/kg) and ketamine (80–100 mg/kg body weight), and their abdomen was cleansed with betadine solution. A small incision, 5 mm in length, was made along the ventral midline of the abdomen, first opening skin followed by access into the peritoneal cavity. The first native group received 6.5 mg of unprocessed native bovine bone substitutes implanted into the peritoneal cavity to serve as a positive control, while the second sham (SO) group of mice did not receive any implant material and served as a negative control. The third group received 6.5mg of DMB granules, and the fourth group received 6.5 mg of DCC granules in the peritoneal cavity. The peritoneal sheath layer and abdominal skin in all animals were sutured using silk suture size 4/0. The animals were monitored regularly during the post-implantation period. Four animals per group were euthanized by cervical dislocation at four time points: days 1, 7, 14 and 21.

Blood samples were collected aseptically and processed to obtain serum. Peritoneal fluid from all the experimental groups and the three abdominal organs—spleen, kidney, and liver—were also collected. The weight of the spleen was recorded, followed by cell isolation and processing for flow cytometric analysis. At the same time, kidney and liver tissue samples were also fixed in 10% neutral buffered formalin for histological assessments.

## Blood count and spleen index

At each time point, blood was collected directly from the heart using a fine needle into collection tubes containing antico-agulants. WBCs were measured using a 3-part hematology analyzer manual technique (Shenzhen, China) calibrated for murine blood cells. Spleens were harvested from mice on days 7, 14, and 21 and weighed. The spleen index was calcu-lated using the given formula: Spleen index (mg/10 g) = (weight of spleen/bodyweight × 10) [18].

## Preparation of mouse splenocytes for immune cell analysis

For the isolation of splenocytes, spleen tissue was minced using fine scalpel blades and forced to pass through a 40 μ filter (BD Biosciences). This would yield single-cell suspension of spleenocytes, which were then suspended in RBC lysis buffer (eBiosciences, USA). The cell suspension was incubated at room temperature for 15 minutes with intermittent mild vortexing every 5 minutes. The cells were washed with PBS and centrifuged at 1200 rpm for 7 minutes to pellet down the spleenocyte population. The cells were then stained with anti-mouse CD3 PerCP-Cy5.5, CD4 Alexa 488, and CD8 APC markers, incubated in the dark for 30 mins, washed using cold PBS, and samples were acquired by using a BD FACSAria III flow cytometer and analysis was performed by using FlowJo software.

## Real-time PCR analysis of cytokines expression

The expression of pro- and anti-inflammatory cytokines in the peritoneal lavage was assessed on day 1 post-implantation. The expression of IL-1β, TNF-α, IL-6, and IL-10 genes was evaluated by real-time PCR. Briefly, total RNA was extracted using RNA Minikit (Qiagen, USA) and concentration was measured using Nano-drop ND1000 (Thermo Scientific, USA). First-strand cDNA was synthesized using a High Script cDNA synthesis kit (Qiagen) and expression of the genes was quantified by real-time RT-PCR analysis using 5X FIREPOL SYBR green master mix (Solisbiodyne, USA). 18S served as an internal control in the reaction. Gene-specific primers used in this study are summarized in Table 1 [19].

## Cytokine bead array

The levels of inflammatory cytokines from peritoneal lavage obtained from the 4 groups were assessed on days 1, 7, 14, and 21 using a cytokine bead array. The BD™ Cytometric Bead Array (CBA) mouse Inflammation Kit (BD Biosciences, USA) was employed in the assay. A total of 50 μl of peritoneal exudate was analyzed using the manufacturer's guidelines, and the ensuing cytokine expression levels were quantitatively determined. The quantification of inflammatory cytokine expression encompassed the assessment of a specified cytokine panel, including proinflammatory cytokines IL-6, MCP-1, IFN-γ, TNF-α, IL-12p70, and the anti-inflammatory cytokine IL-10. Within each assay, three replicates were established for every experimental group.

**Table 1. Sequences of primers used in qPCR.**

| Genes | Primer sequences |
|---|---|
| IL-1β | FP: TGCCACCTTTTGACAGTGATG<br>RP: AAGGTCCACGGGAAAGACAC |
| TNF-α | FP: GTCCCCAAAGGGATGAGGTG<br>RP: AGAGAGAGGTGTGGGAACACT |
| IL-6 | FP: ACCAGTGACTGAAAGACGCA<br>RP: TGGGGGAGGATGTTTGGATG |
| IL-10 | FP: GCTCCAAGACCAAGGTGTCT<br>RP: CGGAGAGAGGTACAAACGAGG |
| 18S | FP: AGAGCGGGTAAGAGAGGTGT<br>RP: GTCGGGGTCCGACAAAACC |

## Histology

The liver and kidney tissue samples harvested from the study groups were fixed in 10% neutral buffered formalin, embedded in paraffin, and five μm sections were cut using a microtome. Sections were stained with hematoxylin and eosin (H&E) (Sigma). The stained sections were examined in an inverted microscope (IX81, Olympus Corporation, Tokyo, Japan) and screened for morphological alterations.

## Statistical analysis

Statistical differences among the experimental groups were studied using GraphPad Prism Prism software (version 9.1). One-way ANOVA followed by multiple comparisons was performed to calculate total WBC and CD3/CD4/CD8 cell counts. Mixed ANOVA (Tukey's multiple comparisons test) was performed to compare mean differences in spleen index and cytokine array analysis. Data were considered statistically significant if $p < 0.05$.

## Results

### Clinical observations

All animals used in the study survived and remained healthy and active throughout the twenty-one-day study period. There were no alterations in feeding habits, and their body weight among the four groups was comparably maintained. All abdominal wounds healed well by day seven, and no self-inflicted injury was observed at the operation site that could modify the healing or postoperative inflammatory responses.

### Effects of DMB and DCC implantation on leukocyte (WBC) counts

On day 7, the native positive control group showed the highest WBC count. At the same time, the Sham negative control group demonstrated the lowest level of WBC (Fig 1). The DMB group also demonstrated a high WBC count compared to DCC, and both are much higher than the Sham. On day 14, while the Sham remained at the same level, the DMB group showed a slight decrease and was comparable to DCC. By Day 21, the WBC count in the Native Control group remained the highest while the count in DCC has (significantly) decreased further compared to the DMB group, to a level similar to the Sham. WBC level in the DCC group was significantly lower than the native group and DMB group throughout twenty-one days ($p < 0.0001$).

### Spleen index following DMB and DCC implantation

On day 7, retrieval of the abdominal organs from Group 1 mice revealed a clinically obvious large spleen compared to the Sham negative control which appear of normal size. The clinical size of the spleen from the DMB group is approximately similar to the DCC group, but both were much larger when compared with the Sham group. On day 7, the average weight of the spleens from the native bone group was significantly higher than those of the Sham ($p < 0.01$), DMB ($p < 0.05$), and DCC ($p < 0.001$) groups (Fig 2). Similarly, the weight of spleens from the native bone group remained significantly high compared to the other three groups when retrieved on day 14 (Sham ($p < 0.001$), DMB ($p < 0.01$)) and DCC ($p < 0.001$) and day 21 (Sham ($p < 0.01$), DMB ($p < 0.01$) and DCC ($p < 0.01$) respectively. The Sham group recorded the lowest spleen weight while there was a noticeable increasing trend in the average spleen weight for both the DMB and DCC groups compared to the Sham group across all time points; however, the data was not statistically significant. Likewise, between DMB and DCC groups, DMB exhibited a higher average spleen weight than DCC, but this difference was also not statistically significant (Fig 2).

### Effects of DMB and DCC implantation on CD4[+] and CD8[+] cell counts

CD4[+] and CD8[+] cell count in the spleen was used for assessing the immunotoxicity of DMB and DCC substitute implantation. At day 1 post-implantation, the splenocytes from Native groups demonstrated a significantly higher proportion of

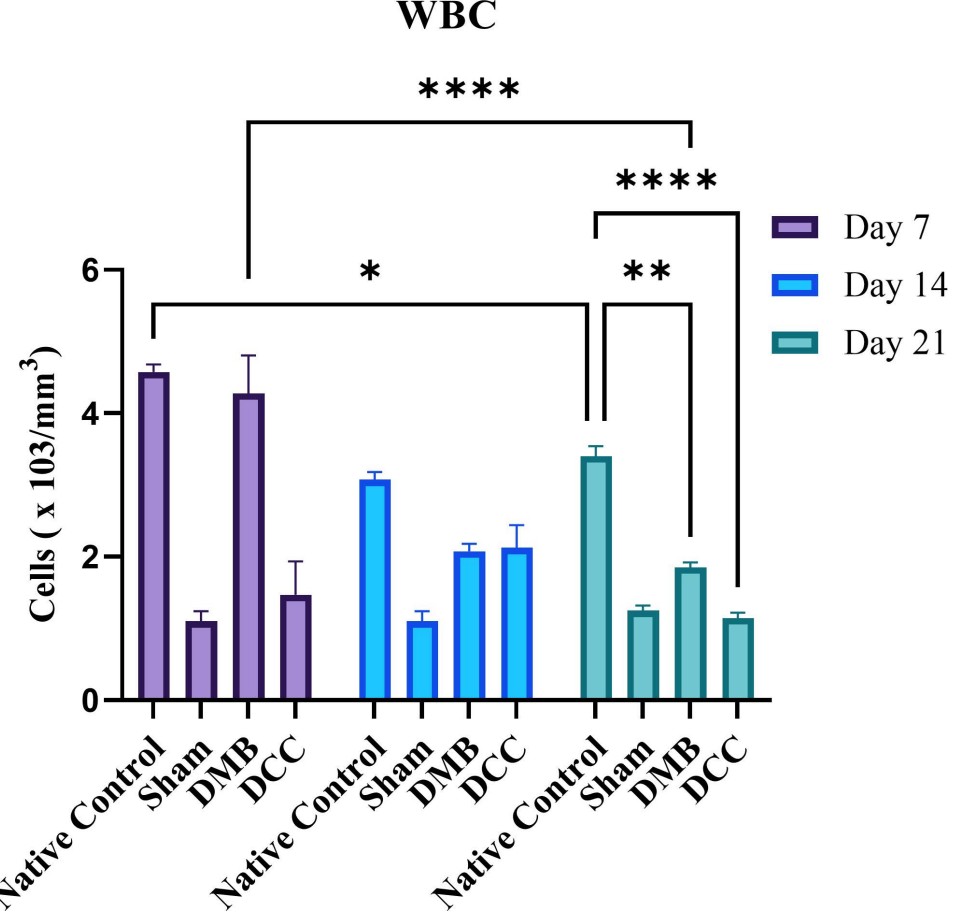

**Fig 1. Analysis of WBC count.** BALB/c mice were implanted with unprocessed native bovine bone (positive control), DMB and DCC substitutes and a sham operation (negative control). WBC count was analyzed in all groups at days 7, 14, and 21 post-implantations. n = 4 animals/group. *p < 0.05, **p < 0.001, and ***p < 0.0001 are indicated.

CD3+ cells compared to the DMB (p < 0.0001) and DCC (p < 0.05) groups. At the same time point, the DCC group demonstrated significantly higher CD3+ cell counts compared to the DMB group (p < 0.0001) ([Fig 3A](), [3B]() and [S2 Fig]()).

On day 1, the DCC group showed significantly lower CD4+ T-cell count compared to both the native (p < 0.001) and DMB (p < 0.05) groups, while the proportion of cytotoxic CD8+ cells in DCC was significantly higher compared to the DMB (p < 0.0001) and native group (p < 0.0001). The CD4+/CD8+ count ratio did not show significant variations between the native and DMB groups, while their ratio was significantly lower in the DCC group (p < 0.05) ([Fig 3A](), [B]()).

On day 7, the Group 1 native bone group demonstrated the highest levels of CD3+ T-cells compared to the other implanted groups, although the data was not significant. Additionally, the DMB group on Day 7 and beyond showed significantly higher CD3+ cells compared to the DCC group (p < 0.0001), and the CD3+ cell count was consistently lowest in the DCC group compared to DMB and Native group across all time points. Still, the ratio of CD4+/CD8+ cells did not exhibit any significant differences among the three implanted groups at this time point ([Fig 3B]() and [S2 Fig]()).

As the duration of implantation progressed to day 14, the levels of T cell subsets decreased further in the DMB and DCC groups. Notably, by day 21, the native group maintained the highest T-cell levels, while T-cell subsets of CD4+ and CD8+ were comparable in all groups and maintained an equivocal CD4+/CD8+ ratio.

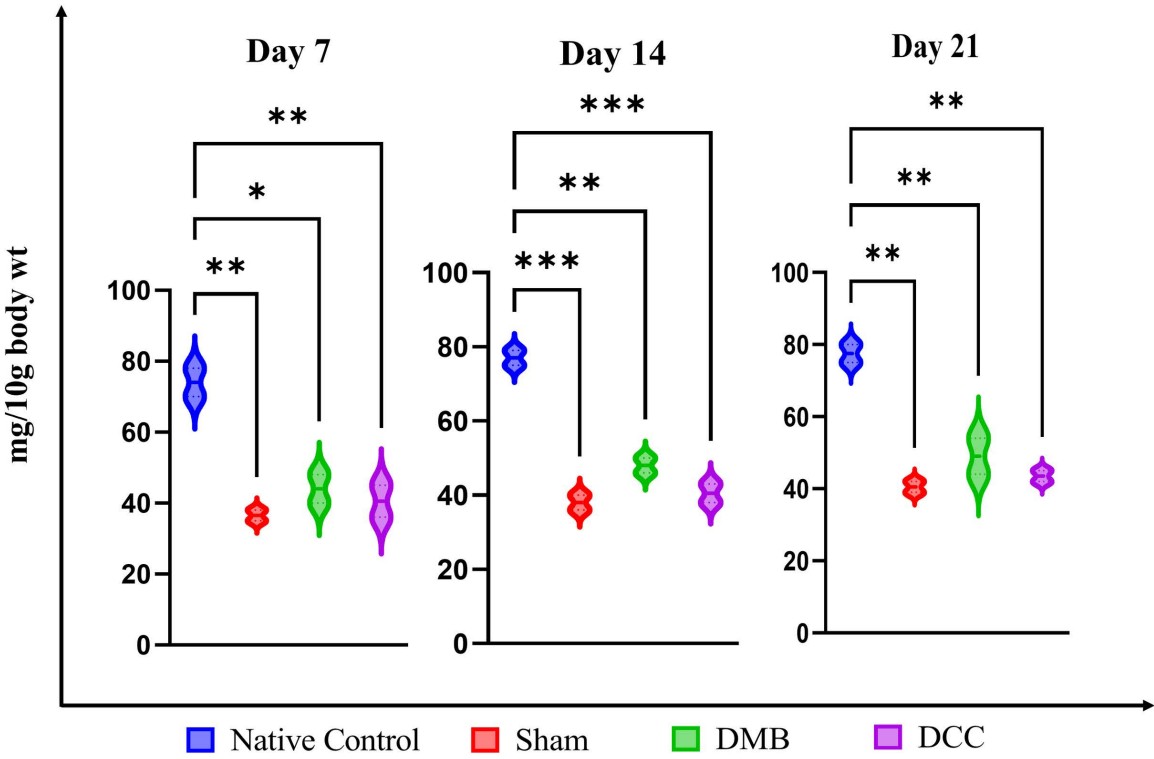

**Fig 2. Spleen index.** Analysis of average spleen weights and spleen index of BALB/c mice implanted with unprocessed native bovine bone (positive control), Sham Operation (negative control), implantation with DMB, and DCC substitutes at days 7, 14 and 21. n = 4 animals/group. *p < 0.05, **p < 0.01, and ***p < 0.001 are indicated.

### Gene expression analysis of pro- and anti-inflammatory cytokines in peritoneal fluid cells

Fig 4 presents the results of real-time PCR analysis, showing the expression of pro- and anti-inflammatory cytokines in the peritoneal fluid after one day of implantation of DMB and DCC substitutes, along with native controls. DMB-treated animals exhibited a significant increase in mRNA for pro-inflammatory cytokines such as IL-1β, TNF-α, and IL-6 (p < 0.0001) compared to the DCC group. Similarly, the anti-inflammatory cytokine, IL-10 expression was also found to be significantly higher in the DMB group (p < 0.0001) compared to the DCC (Fig 4).

### Cytokine and chemokine expression of peritoneal fluid

Expression of cytokines and chemokines from peritoneal fluid was significantly higher in the native group compared to the DMB and DCC groups (Fig 5). Their levels declined abruptly and sequentially from day 1 to day 21. Significant increases in IL-12, TNF-α, IFN-γ, MCP-1, and IL-6 were evident on day one post-implantation in all the groups. Subsequent measurements in later days showed a decline in these elevated cytokine and chemokine levels. The anti-inflammatory cytokine IL-10 also showed a similar pattern of increased levels on day one, and it was significantly higher in the native group (p < 0.0001) compared to DMB and DCC groups and remained high on days 7 and 14 (p < 0.001). The level of IL-10 in DMB and DCC was comparable on days 14 and 21.

### Histological evaluation of liver and kidney tissues

On day 7, histological examination of liver tissue from the native group displayed abundant pyknotic nuclei (condensed chromatin) and necrotic cells with mild vacuolation (Fig 6). Pynokic cell presence was also observed in the native group

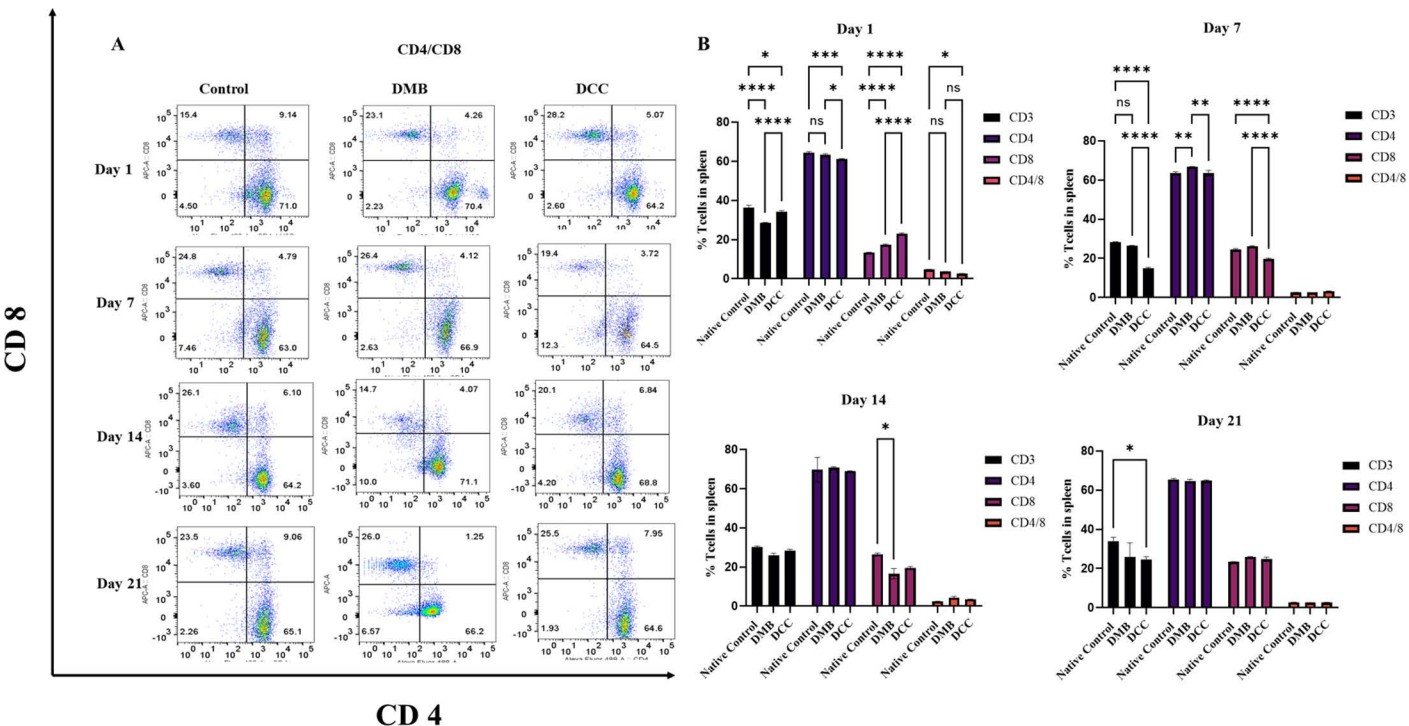

**Fig 3. Flow cytometric analysis of CD4 $^+$and CD8$^+$cells.** CD4$^+$ and CD8$^+$ cells in splenocytes following implantation of unprocessed native bovine bone (positive control), sham operation (negative control), and implantation of DMB, and DCC substitutes at days 7, 14 and 21. **(A)** Flow cytometry dot plot representing the % of CD4$^+$ and CD8$^+$ T-cells across the different time points. **(B)** Bar graphs representing the proportion of CD3$^+$, CD4$^+$, CD8$^+$, and the ratio of CD4/CD8 cells at days 7, 14 and 21. n = 4 animals/group. *p < 0.05, **p < 0.001, and ***p < 0.0001 are indicated.

on day 14, albeit to a lesser extent and with a mild level of lymphocyte infiltration. By day 21, such changes in the nuclear level were not evident.

In the DMB and DCC groups, histological examination of liver samples displayed normal liver morphology at all time points. Hepatocytes with large polygonal cells having round nuclei, prominent nucleoli, and pink cytoplasm were present. The liver parenchyma, which comprises thousands of small, roughly hexagonal lobules, was similar to that of the Sham group (Fig 6).

At day 7, histological examination of kidney tissue from the native bone group displayed increased vascularity with hemorrhage, evidenced by the extravasation of RBCs (Fig 7). The acute inflammatory histological features remained the same at day 14. In contrast, by day 21, chronic inflammatory responses were demonstrated by dilatation of vessels and extravasation of erythrocytes persisted in the renal medulla. Morphological features of the glomerulus and kidney tubule cells in the native bone group were seen intact throughout. On day 7, histological examination of kidney samples in DMB and DCC groups showed mild extravasated erythrocytes with intact glomeruli and kidney tubules seen in both groups displaying normal morphology at all time points similar to the Sham group.

## Discussion

In immunology testing for biomaterials, the immune response can vary significantly depending on the site of implantation. Implanting a biomaterial in different tissues like the peritoneum or bone will lead to different immunological outcomes due to the distinct microenvironments of these sites. In this study, the peritoneal model was chosen since this cavity is a site rich in immune cells, particularly macrophages, dendritic cells, and neutrophils, and it is part of the body's primary defense

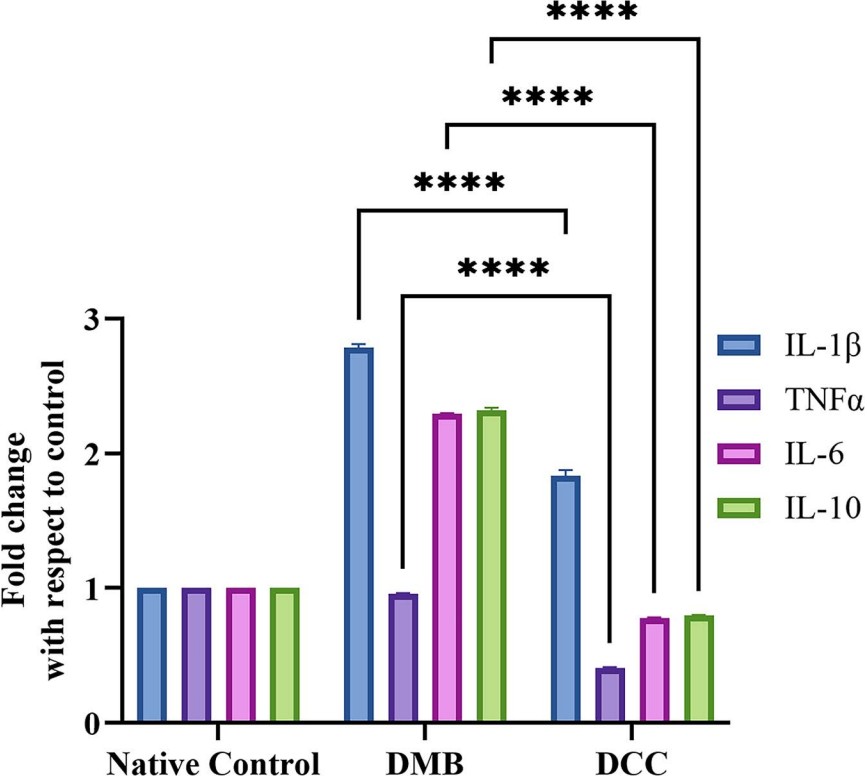

**Fig 4. Real-time PCR expression of IL1- β, TNF-α, IL-6, and IL-10 cytokines in the peritoneal fluid extracted from BALB/c mice following implantation of unprocessed native bovine bone (positive control), DMB, and DCC substitutes at day 1.** The data was normalized using the native control group and 18S served as the internal housekeeping gene. ****p < 0.0001 is indicated.

system against foreign invaders. Implanting the bone substitutes in the peritoneum generally leads to a rapid immune response [20,21]. Thus, this study evaluated the rather rapid short-term immunological responses of DMB and DCC in the peritoneal model over a 21-day period.

An increase in WBC counts in the native, DMB, and DCC groups from Day 1 aligns with the rise in proinflammatory cytokines in the peritoneal fluid. This early immune activation provides insight into how different bone substitutes modulate the inflammatory response at the molecular level. Gene expression analysis revealed that DMB had significantly higher levels of IL-1β, TNF-α, IL-6, and the anti-inflammatory cytokine IL-10 compared to DCC on Day 1, suggesting a stronger early immune response. Cytokine profiling at the protein level also showed activation of pro-inflammatory cytokines such as IL-12, TNF-α, IFN-γ, MCP-1, and IL-6, particularly on Day 1. The reduced IFN-γ response in DMB and DCC compared to native bone suggests a lower risk of acute rejection [22,23].

Notably, IL-12 was sustained in both native bone and DMB groups over the 3-week period. Increased IL-12 expression is known to escalate inflammatory responses by recruiting Th1 lymphocytes at the implant sites. This aligns with our previous findings [24], where DMB elicited a greater immune cell response than DCC, favouring Th1 cytokine production. Although the humoral component plays a crucial role, the rejection of bone grafts is considered cellular [25]. Therefore, this study further performed systemic immune compatibility measurements by CD4(+) and CD8(+) cell counts in the spleen. The higher CD3 + T-cell levels in the native and DMB groups compared to DCC at day one imply a robust immune activation. The significantly lower CD4⁺ T-cells and increased CD8⁺ T-cells in the DCC group at day one compared to both native and DMB groups suggest an early cytotoxic response specific to DCC. However, their levels declined by day seven.

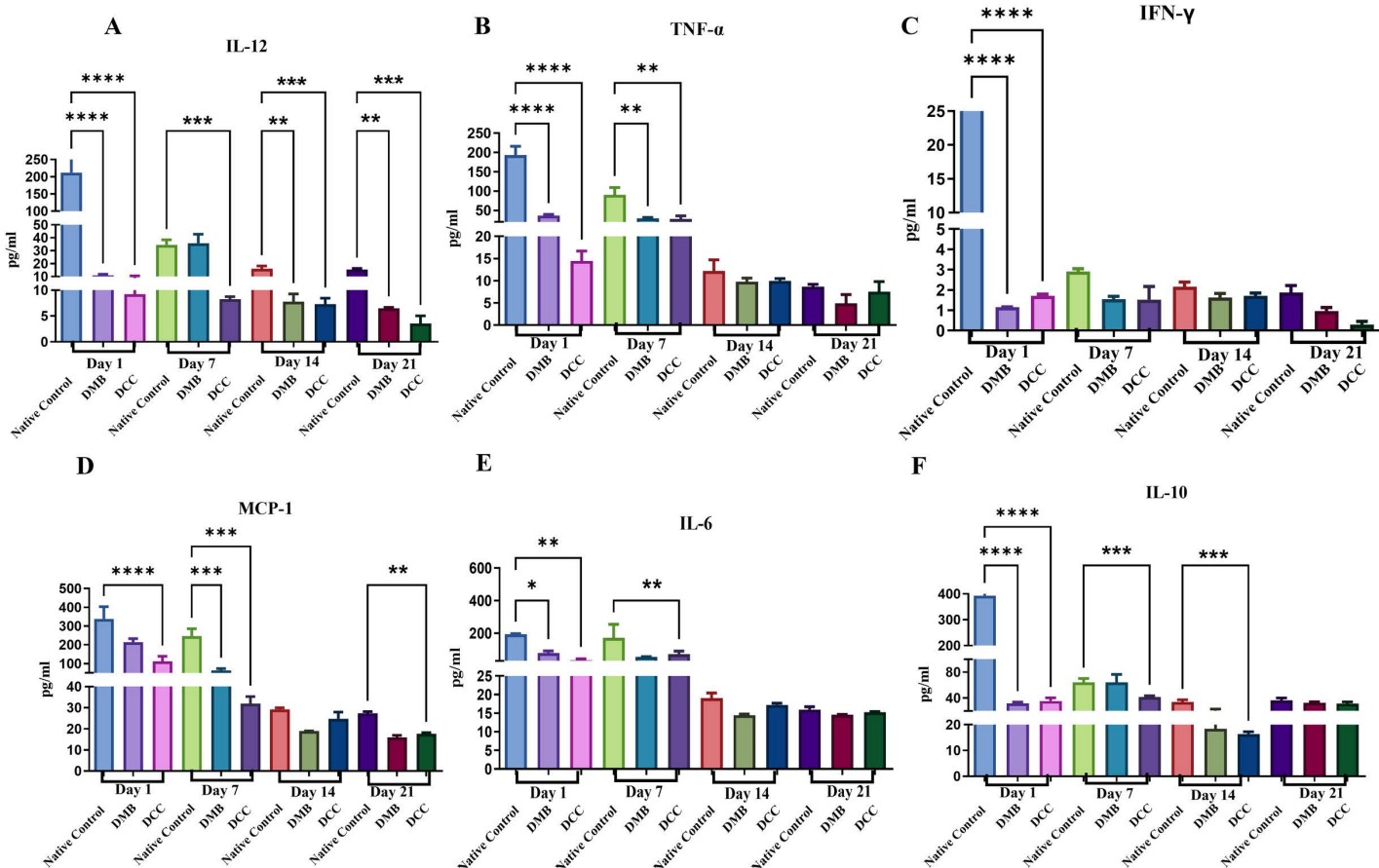

**Fig 5. Cytokine profiling of peritoneal fluid extracted from BALB/c mice following implantation of unprocessed native bovine bone (positive control), DMB, and DCC substitutes at days one, seven, fourteen and twenty-one.** The levels of cytokines were normalized using a sham operation (negative control) group. n = 4 animals/group. *p < 0.05, **p < 0.001, and ***p < 0.0001 are indicated.

The heightened proinflammatory response in DMB in comparison to DCC suggests an underlying mechanism beyond cytokine upregulation, potentially involving endogenous danger signals such as Damage-Associated Molecular Patterns (DAMPs). DAMPs, including HMGB1 and S100 proteins, are released from necrotic cells [26], or ECM fragmentation, triggering immune activation [27,28]. These molecular signals can interact with immune receptors, triggering proinflammatory cascades. In the demineralization process, the removal of mineral content exposes organic matrix components, growth factors, and other molecules, which may trigger immune reactions [29,30]. Therefore, the processing methods of the biomaterials can influence their immunological responses, with DMB processing potentially contributing to the increased inflammatory response through DAMP activation [31].

We observed an increase in spleen size and weight in the DMB and DCC groups compared to the Sham group, an indicator of the immuno-toxicity in the native bone group, signifying heightened splenic immune activity due to exposure to unprocessed xenogeneic bone substitutes [32]. The enlargement and increased weight of the liver and spleen (hepatosplenomegaly) following the implantation of bovine bone substitutes in the peritoneum of mice is typically a response to systemic immune activation and inflammation. The liver is central to the body's immune response, acting as a site for antigen processing and cytokine production. It contains a large number of immune cells, such as Kupffer cells (liver macrophages), which can become activated and lead to inflammation and hypertrophy (enlargement) of the liver. The

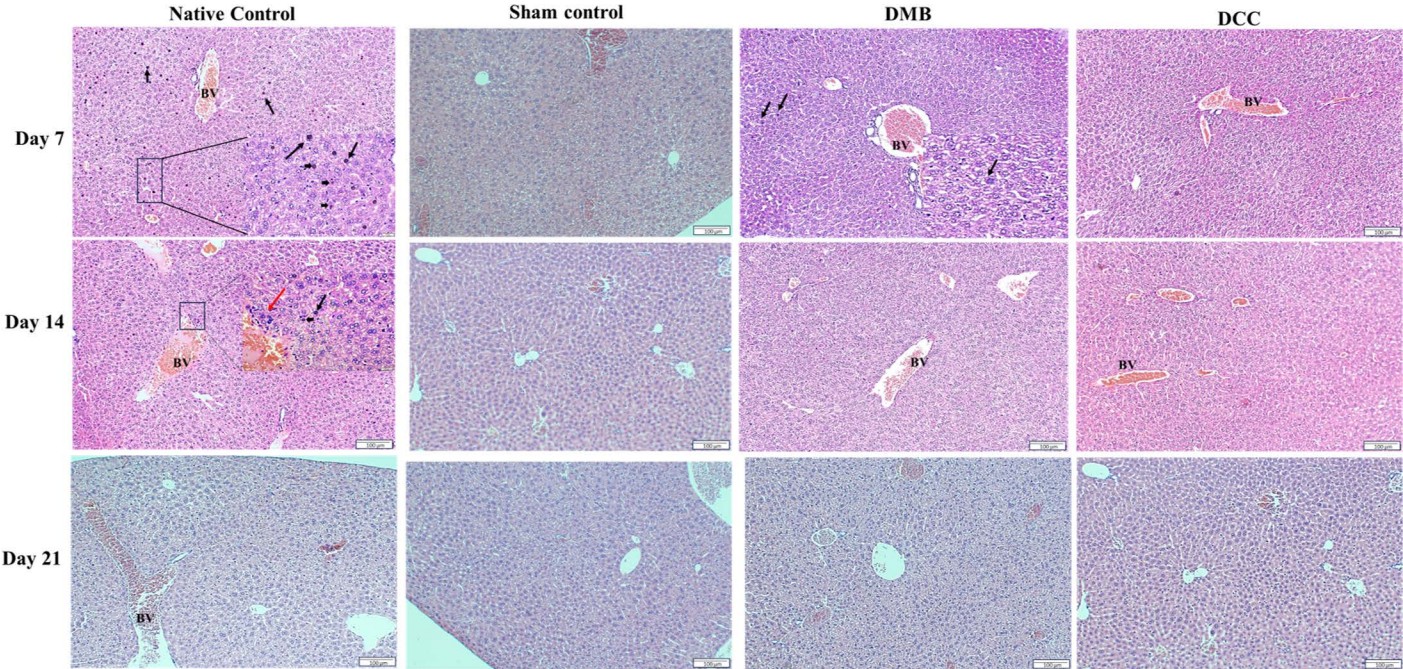

**Fig 6. Histology of liver tissues in BALB/c mice following implantation of unprocessed native bone positive control, sham Negative control, DMB, and DCC substitutes at days 7, 14, and 21 by H&E staining.** The abundant presence of deeply stained (blue) pyknotic nuclei and necrotic cells with mild vacuolation (black arrowheads) is seen in the native bone control group on days 7 and 14. The native bone group at day 14 showed mild infiltration of lymphocytes (red arrows; inset). Black arrows indicate deeply stained pyknotic nuclei, arrowheads indicate necrotic cells with mild vacuolation, and blood vessels (BV) are shown. Scale bar = 100 μm (main images); scale bar = 20 μm (inset images). n = 4 animals/group.

spleen is a major organ involved in immune responses, filtering blood and serving as a reservoir for immune cells such as lymphocytes and macrophages. Upon exposure to foreign antigens, the spleen can become enlarged due to an increase in immune cell proliferation (splenomegaly) [33,34]. In the present study, both DMB and DCC processing techniques reduced splenic immune activity significantly compared to the native bone group.

To further validate the systemic immune response findings, histological assessment was conducted on abdominal organs. Histological analysis of abdominal organs consistently corroborated the hematological and cytokine profile, with the native bone group exhibiting significant inflammatory cell infiltration and kidney tissue damage. In contrast, the DMB and DCC groups showed no marked renal morphology differences, suggesting the material's safety. These results align with previous studies that have demonstrated the benefits of decellularization in reducing graft immunogenicity and promoting tissue integration [35–37].

In this study, all animals survived and remained active until the end of the implantation period, suggesting that both DMB and DCC induce a low inflammatory response that does not reach a lethal threshold. When this immunological data was compared to other common biomaterials such as hydroxyapatite, a wide range of biocompatibility results were observed since the foreign body reaction to biomaterials also depends on the particle size, shape, topography, and chemical composition and does not allow accurate comparisons when implanted in different body sites. While implantation studies of hydroxyapatite in bone defects tend to elicit a lower immune response because calcified structure is less prone to significant immune cell infiltration, similar material when implanted into soft tissue tends to provoke a much higher inflammatory response due to the more vascular and cellular soft tissue environment [23,38,39]. Taken together, the current study illustrates the differential immune responses of a natural bone substitute subjected to different physicochemical processes.

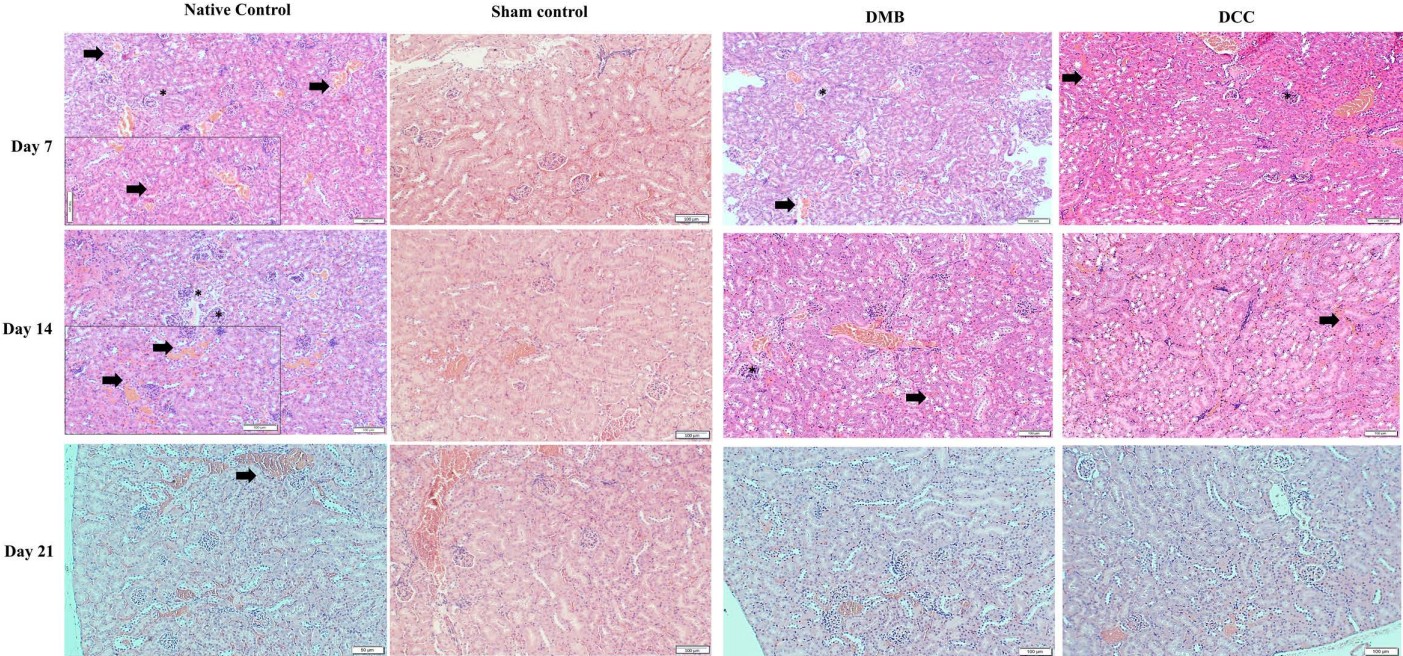

**Fig 7. Histology of kidney tissues in BALB/c mice following implantation of unprocessed native positive control bone, Sham Negative Control, DMB, and DCC substitutes at day 7, 14, and 21 by H&E staining.** The native control at day seven showed abundant erythrocyte extravasation (black arrowhead) and increased vascularity. Normal glomerulus (*) and tubule cells are seen. DMB and DCC groups showed mild extravasated erythrocytes with intact glomeruli and kidney tubules. Epithelial cells containing nuclei (dark spots) are well preserved in the native bone control group, DMB, and DCC implanted groups. Scale bar = 100 μm (main images); scale bar = 20 μm (inset images). n = 4 animals/group.

There are obvious limitations to our results under the current experimental conditions since this study follows immune responses only up to 21 days, which may not capture the full spectrum of chronic immune reactions. Hence, extending the study duration to assess long-term immune responses and potential tissue integration is warranted.

## Supporting information

**S1 Fig.** Granular form of bovine bone, DMB "Demineralized bone" and DCC "Decellularized bone"
(TIF)

**S2 Fig.** Flow cytometric analysis of CD3+cells in spleens in native control bone, DMB, and DCC granules implanted mice at days 1, 7, 14, and 21.
(TIF)

## Acknowledgments

The authors expressed their gratitude to Ms. Razaz Omer of the Research Institute for Medical and Health Sciences for her contribution to Histological analyses.

## Author contributions

**Conceptualization:** K. G. Aghila Rani, Ali Al Qabbani, Sausan AlKawas, Ahmad Azlina.

**Data curation:** K. G. Aghila Rani, Ali Al Qabbani, Ahmad Azlina.

**Investigation:** K. G. Aghila Rani, Manju Nidagodu Jayakumar, Suzina S. A. H., A. R. Samsudin.

**Methodology:** K. G. Aghila Rani, Ali Al Qabbani, Manju Nidagodu Jayakumar, Suzina S. A. H., A. R. Samsudin, Ahmad Azlina.

**Project administration:** Sausan AlKawas, A. R. Samsudin.

**Resources:** Manju Nidagodu Jayakumar, Suzina S. A. H.

**Supervision:** Ali Al Qabbani, Sausan AlKawas, A. R. Samsudin.

**Validation:** K. G. Aghila Rani, Ali Al Qabbani, Sausan AlKawas, Manju Nidagodu Jayakumar, A. R. Samsudin, Azlina Ahmad.

**Writing – original draft:** K. G. Aghila Rani, Ali Al Qabbani, A. R. Samsudin, Ahmad Azlina.

**Writing – review & editing:** Ali Al Qabbani, A. R. Samsudin, Ahmad Azlina.

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
