## [Decision Letter · Decision Letter 0]

2 Feb 2025

PONE-D-25-01024A COMPARATIVE STUDY ON IMMUNE RESPONSES TO DEMINERALIZED AND DECELLULARIZED BONE SUBSTITUTE FOLLOWING INTRAPERITONEAL IMPLANTATION IN MOUSE MODELPLOS ONE

Dear Dr. Azlina,

Thank you for submitting your manuscript to PLOS ONE. After careful consideration, we feel that it has merit but does not fully meet PLOS ONE’s publication criteria as it currently stands. Therefore, we invite you to submit a revised version of the manuscript that addresses the points raised during the review process.

The manuscript has been carefully evaluated by two experts in the field. Both found the manuscript of interest pending some major amendments that are needed in order to reduce the fact that is rather descriptive. Moreover some molecular aspects must be also analyzed.This Authors must follow all the indications specified by both the referees.==============================

We look forward to receiving your revised manuscript.

Kind regards,

Gianpaolo Papaccio, M.D., Ph.D.

Academic Editor

PLOS ONE

Journal Requirements:

2. To comply with PLOS ONE submissions requirements, in your Methods section, please provide additional information regarding the experiments involving animals and ensure you have included details on (1) methods of sacrifice, and (2) efforts to alleviate suffering.

“The authors expressed their gratitude to Ms. Razaz Omer of the Research Institute for Medical and Health Sciences for her contribution to Histological analyses. We would also like to acknowledge the funding resources from the University of Sharjah, UAE (grant no. 23001100169) awarded to Prof. Rani Samsudin.”

“We would also like to acknowledge the funding resources from the University of Sharjah, UAE (grant no. 23001100169) awarded to Prof. Rani Samsudin.”

Reviewers' comments:

Reviewer's Responses to Questions

**Comments to the Author**

1. Is the manuscript technically sound, and do the data support the conclusions?

Reviewer #1: Yes

Reviewer #2: Partly

2. Has the statistical analysis been performed appropriately and rigorously? 

Reviewer #1: Yes

Reviewer #2: Yes

3. Have the authors made all data underlying the findings in their manuscript fully available?

Reviewer #1: Yes

Reviewer #2: Yes

4. Is the manuscript presented in an intelligible fashion and written in standard English?

Reviewer #1: Yes

Reviewer #2: No

5. Review Comments to the Author

Reviewer #1: In this study Authors investigate the immune responses following implantation of lyophilized demineralized (DMB) and lyophilized decellularized (DCC) bovine cancellous bone substitutes in a mouse peritoneal model. It assesses immunogenicity through hematological parameters, cytokine profiling, immune cell populations, and histological examination. The findings suggest that both DMB and DCC demonstrate favorable immunocompatibility, with DCC exhibiting superior immune tolerance.

This manuscript presents valuable findings on the immunogenicity of bone substitutes and is well-structured with rigorous experimental methods. However, it has some limitations, particularly regarding the small sample size, lack of long-term follow-up, and limited mechanistic insights.

More in depth:

The rationale for using an intraperitoneal implantation model rather than an orthotopic (bone defect) model is not well explained. Authors must provide a clearer justification for why this model was chosen over a direct bone implantation model, which may more accurately reflect clinical applications.

The study follows immune responses only up to 21 days, which may not capture the full spectrum of chronic immune reactions. Authors must consider extending the study duration to assess long-term immune responses and potential tissue integration.

The study lacks mechanistic exploration, particularly in detailing how DCC achieves better immunocompatibility at the molecular level. Authors must include gene expression analyses, to further elucidate pathways responsible for immune modulation.

Authors must include comparative data or discuss how others bone graft materials, such as hydroxyapatite or allografts, perform relative to current clinical alternatives.

Reviewer #2: This study aimed to examine the immune responses of lyophilized demineralized (DMB) and lyophilized decellularized (DCC) bovine cancellous bone substitutes following implantation into the peritoneal cavity of BALB/c mice over a 21-day observation period. To assess the immunological responses and immunocompatibility of those bone substitutes, hematological assessments, spleen index measurements, immune cell responses, and cytokine profiling were performed, along with the histological examination of the spleen, liver, and kidney.

The manuscript is interesting and is well-organized, featuring robust experimental methods. Although this, there are several concerns that need to be addressed. The introduction and discussion are too long, they must be reduced. In particular, the discussion is redundant and must focus on results, novelty and limitations of the study. So, it must be completely revised. The authors should clarify why they chose the intraperitoneal implantation model. Being the manuscript merely descriptive, the authors must explore the molecular mechanism by which DCC bone substitutes promote superior immunocompatibility compared to DMB. The main text makes no reference to supplementary figure 2. Moreover, there are no legends of supplementary figures.

6. PLOS authors have the option to publish the peer review history of their article (what does this mean? ). If published, this will include your full peer review and any attached files.

**Do you want your identity to be public for this peer review?** For information about this choice, including consent withdrawal, please see our Privacy Policy .

Reviewer #1: No

Reviewer #2: No

---

## [Author Response · Author response to Decision Letter 1]

27 Mar 2025

Editor’s Comments Responses

Manuscript meets PLOS ONE’s style requirements

2. To comply with PLOS ONE submission requirements, in your Methods section, please provide additional information regarding the experiments involving animals and ensure you have included details on (1) methods of sacrifice, and (2) efforts to alleviate suffering.

Highlighted in page 6 & 7

Added and corrected in “Funding information section”.

“The authors expressed their gratitude to Ms. Razaz Omer of the Research Institute for Medical and Health Sciences for her contribution to Histological analyses. We would also like to acknowledge the funding resources from the University of Sharjah, UAE (grant no. 23001100169) awarded to Prof. Rani Samsudin.”

Thank you very much

“We would also like to acknowledge the funding resources from the University of Sharjah, UAE (grant no. 23001100169) awarded to Prof. Rani Samsudin.”

As kindly advised, we have removed the funding information from the acknowledgement section of the manuscript.

We would like to retain the funding statement as ‘“We would also like to acknowledge the funding resources from the University of Sharjah, UAE (grant no. 23001100169) awarded to Prof. Rani Samsudin.”

We have included the funding information in the cover letter and thank you for updating it in the online submission form.

We adhere to the open data policy and all data is contained within the manuscript.

We shall revise the statement in the online submission form.

Added and highlighted in page 20

Reviewer 1

1. In this study Authors investigate the

immune responses following implantation of lyophilized demineralized (DMB) and lyophilized decellularized (DCC) bovine cancellous bone substitutes in a mouse peritoneal model. It assesses immunogenicity through hematological parameters, cytokine profiling, immune cell populations, and histological examination. The findings suggest that both DMB and DCC demonstrate favorable immunocompatibility, with DCC exhibiting superior immune tolerance.

This manuscript presents valuable findings on the immunogenicity of bone substitutes and is well-structured with rigorous experimental methods. However, it has some limitations, particularly regarding the small sample size, lack of long-term follow-up, and limited mechanistic insights.

Thank you very much for the valuable comments. We appreciate the recognition of the strengths of our study and the constructive feedback provided.

However, the small sample size was due to multiple time points and more than 1 group of comparison, so distribution of numbers of animals was among these different groups, and we adhere to the animal ethics protocol that emphasize using the minimum number of animals that may provide meaningful results.

Highlighted in pages 2,14 & 17.

For limited mechanistic insights, we have now added the gene expression studies for validating the differential expression of pro- and anti-inflammatory cytokines among the study groups that support the results we observed at the protein level (Fig 4 in the revised manuscript).

2. The rationale for using an intraperitoneal implantation model rather than an orthotopic (bone defect) model is not well explained. Authors must provide a clearer justification for why this model was chosen over a direct bone implantation model, which may more accurately reflect clinical applications.

Highlighted and justified in page 17

In immunology testing for bone substitutes, the immune response can vary significantly depending on the site of implantation. Implanting bone substitutes in different tissues like the peritoneum or bone will lead to different immunological outcomes due to the distinct microenvironments of these sites. The peritoneal cavity is a site rich in immune cells, particularly macrophages, dendritic cells, and neutrophils, and it is part of the body’s primary defense system against foreign invaders. Implanting demineralized or decellularized bovine bone substitutes in the peritoneum leads to a rapid immune response, as shown in our study, since the peritoneal cavity is highly vascularized, and immune cells can quickly migrate to the site of implantation. In contrast, implantation of these bone substitutes in in-vivo bone defects may also initiate an immune response, but it often lead to a more delayed response, and less intense compared to implantation in soft tissues like the peritoneum. Thus bone substitutes implantation study in bone defects for shorter duration studies may demonstrate favorable ‘biocompatibility’ while the same material when implanted in the mice may show early immunologic response.

References (23,38,39) in page 20 and in references in pages 24 & 25

3. The study follows immune responses only up to 21 days, which may not capture the full spectrum of chronic immune reactions. Authors must consider extending the study duration to assess long-term immune responses and potential tissue integration.

Thank you very much for this suggestion. We shall consider continuing our study for longer duration in our future experiments to investigate the long-term immune responses.

The 21-day period is used in immunological studies to evaluate acute and early adaptive immune responses. This allows for the assessment of key inflammatory phases, including innate immune activation, early adaptive responses, and initial tissue remodeling. The primary objective of this study is to assess the initial immune reaction, which is crucial in determining early biocompatibility and potential adverse immune responses. While chronic immune responses are important, they are beyond the current study’s scope. However, future studies can build upon these findings to investigate long-term immune responses and tissue integration.

4. The study lacks mechanistic exploration, particularly in detailing how DCC achieves better immunocompatibility at the molecular level. Authors must include gene expression analyses, to further elucidate pathways responsible for immune modulation.

We have explored and hypothesized the possible mechanisms of how DCC achieved better immunocompatibility at the molecular level through several previous experiments:

1. The residual nucleic acid in DCC is much lower than DMC thus promoting a much lower immunogenic response when exposed to human osteoblasts invitro.

2. DMB provoke a stronger immunologic response compared to DCC when exposed to human peripheral blood monocyte-derived macrophages (PBMM) suggesting decellularization process of tissues dampen down inflammatory reaction when exposed to PBMM.

References:

i) Al Qabbani A, Rani KGA, Syarif J, AlKawas S, Sheikh Abdul Hamid S, Samsudin AR, et al. (2023) Evaluation of decellularization process for developing osteogenic bovine cancellous bone scaffolds in-vitro. PLoS ONE 18(4): e0283922. https://doi.org/10.1371/journal.pone.0283922

ii) Rani KGA, Al-Rawi AM, Al Qabbani A, AlKawas S, Mohammad MG, Samsudin AR (2024) Response of human peripheral blood monocyte derived macrophages (PBMM) to demineralized and decellularized bovine bone graft substitutes. PLoS ONE 19(4): e0300331. https://doi.org/10.1371/journal.pone.0300331

Additional gene expression study has been included in the result section pages 9 & 14 respectively & Fig 4

5. Authors must include comparative data or discuss how others bone graft materials, such as hydroxyapatite or allografts, perform relative to current clinical alternatives.

In this study, survival of all animals that remain active till the end of the implantation period suggest that both DMB and DCC excite a low inflammatory response that does not mount to lethal dose. When this immunological data was compared to other common biomaterials such as hydroxyapatite, a wide range of biocompatibility results were observed since the foreign body reaction to biomaterials also depend on the particle size, shape, topography and chemical composition and does not allow accurate comparisons when implanted in different body sites. While implantation studies of hydroxyapatite in bone defects tend to elicit a lower immune response because calcified structure is less prone to significant immune cell infiltration, similar material when implanted into soft tissue tend to provoke a much higher inflammatory response due to the more vascular and cellular soft tissue environment.

References:

1) Li Liu, Hao Chen, Xue Zhao et al. Advances in the application and research of biomaterials in promoting bone repair and regeneration through immune modulation. Materials Today Bio 30 (2025) https://doi.org/10.1016/j.mtbio.2024. 101410

2) James M. Anderson, Analiz Rodriguez and David T. Chang. Foreign body reaction to biomaterials. Seminars in Immunology 20 (2008) 86–100. doi:10.1016/j.smim.2007.11.004

3) Ruhe, P. Q., Hedberg, E. L., Padron, N. T., Spauwen, P. H. M., Jansen, J. A., & Mikos, A. G. (2005). "Biocompatibility and degradation of poly(DL-lactic-co-glycolic acid)/calcium phosphate cement composites. "Journal of Biomedical Materials Research Part A, 74A(4), 533-544.

DOI: 10.1002/jbm.a.30318

As kindly suggested, we have included this information in the discussion of the revised manuscript (page 20).

Reviewer 2

1. This study aimed to examine the immune responses of lyophilized demineralized (DMB) and lyophilized decellularized (DCC) bovine cancellous bone substitutes following implantation into the peritoneal cavity of BALB/c mice over a 21-day observation period. To assess the immunological responses and immunocompatibility of those bone substitutes, hematological assessments, spleen index measurements, immune cell responses, and cytokine profiling were performed, along with the histological examination of the spleen, liver, and kidney.

Thank you very much

2. The manuscript is interesting and is well-organized, featuring robust experimental methods. Although this, there are several concerns that need to be addressed. The introduction and discussion are too long, they must be reduced. In particular, the discussion is redundant and must focus on results, novelty and limitations of the study. So, it must be completely revised.

Thank you very much for the comments, The introduction and discussion have been revised as advised and shortened accordingly.

3. The authors should clarify why they chose the intraperitoneal implantation model. Being the manuscript merely descriptive, the authors must explore the molecular mechanism by which DCC bone substitutes promote superior immunocompatibility compared to DMB. The main text makes no reference to supplementary figure 2. Moreover, there are no legends of supplementary figures.

Highlighted and justified in page 17

In immunology testing for biomaterials, the immune response can vary significantly depending on the site of implantation. Implanting a biomaterial in different tissues like the peritoneum or bone will lead to different immunological outcomes due to the distinct microenvironments of these sites. In this study, the peritoneal model was chosen since this cavity is a site rich in immune cells, particularly macrophages, dendritic cells, and neutrophils, and it is part of the body’s primary defense system against foreign invaders. Implanting the bone substitutes in the peritoneum generally leads to a rapid immune response while testing them in immune-privileged sites such as in bone may generate minimal response and reflect a false impression of immune-compatibility.

Supplementary Fig 2 is referenced in page 13 of main text.

Legends of the supplementary figure have been added and highlighted at the end of the manuscript (page 20).

---

## [Decision Letter · Decision Letter 1]

13 Apr 2025

A COMPARATIVE STUDY ON IMMUNE RESPONSES TO DEMINERALIZED AND DECELLULARIZED BONE SUBSTITUTE FOLLOWING INTRAPERITONEAL IMPLANTATION IN MOUSE MODEL

PONE-D-25-01024R1

Dear Dr. Azlina,

We’re pleased to inform you that your manuscript has been judged scientifically suitable for publication and will be formally accepted for publication once it meets all outstanding technical requirements.

Kind regards,

Gianpaolo Papaccio, M.D., Ph.D.

Academic Editor

PLOS ONE

Additional Editor Comments (optional):

Reviewers' comments:

Reviewer's Responses to Questions

**Comments to the Author**

1. If the authors have adequately addressed your comments raised in a previous round of review and you feel that this manuscript is now acceptable for publication, you may indicate that here to bypass the “Comments to the Author” section, enter your conflict of interest statement in the “Confidential to Editor” section, and submit your "Accept" recommendation.

Reviewer #1: All comments have been addressed

Reviewer #2: All comments have been addressed

2. Is the manuscript technically sound, and do the data support the conclusions?

Reviewer #1: (No Response)

Reviewer #2: Yes

3. Has the statistical analysis been performed appropriately and rigorously? 

Reviewer #1: (No Response)

Reviewer #2: Yes

4. Have the authors made all data underlying the findings in their manuscript fully available?

Reviewer #1: (No Response)

Reviewer #2: Yes

5. Is the manuscript presented in an intelligible fashion and written in standard English?

Reviewer #1: (No Response)

Reviewer #2: Yes

6. Review Comments to the Author

Reviewer #1: (No Response)

Reviewer #2: (No Response)

7. PLOS authors have the option to publish the peer review history of their article (what does this mean? ). If published, this will include your full peer review and any attached files.

**Do you want your identity to be public for this peer review?** For information about this choice, including consent withdrawal, please see our Privacy Policy .

Reviewer #1: No

Reviewer #2: No

---

## [Editor Report · Acceptance letter]

PONE-D-25-01024R1

PLOS ONE

Dear Dr. Azlina,

I'm pleased to inform you that your manuscript has been deemed suitable for publication in PLOS ONE. Congratulations! Your manuscript is now being handed over to our production team.

Kind regards,

on behalf of

Prof. Gianpaolo Papaccio

Academic Editor

PLOS ONE